# Lived experiences of women with low birth weight infants in the Solomon Islands: A descriptive qualitative study

**Lydia S. Kaforau**[1,2]*, **Gizachew A. Tessema**[1,3], **Hugo Bugoro**[2], **Gavin Pereira**[1,4,5], **Jonine Jancey**[1,5]

**1** Curtin School of Population Health, Curtin University, Perth, Australia, **2** School of Nursing and Allied Health Sciences, Solomon Islands National University, Honiara, Solomon Islands, **3** School of Public Health, The University of Adelaide, South Australia, Australia, **4** Centre for Fertility and Health (CeFH), Norwegian Institute of Public Health, Oslo, Norway, **5** enAble Institute, Curtin University, Perth, Australia

* l.kaforau@postgrad.curtin.edu.au

**Data Availability Statement:** The data has been deposited in the qualitative data repository with the link below: https://data.qdr.syr.edu/dataset.xhtml?persistentId=doi:10.5064/F6EIQLEB.

## Abstract

Every year, around 20 million women worldwide give birth to low birth weight (LBW) infants, with majority of these births occurring in low-and middle-income countries, including the Solomon Islands. Few studies have explored the pregnancy lived experience of women who deliver LBW infants. The aim of the study is to understand the lived experience of women in the Solomon Islands who gave birth to LBW infants by exploring their personal (socio-demographic and health), behavioural, social and environmental contexts. We used a qualitative descriptive approach and purposely selected 18 postnatal women with LBW infants in the Solomon Islands for an in-depth interview. All data were analysed using thematic analysis in NVivo. We identified six themes reported as being related to LBW: health issues, diet and nutrition, substance use, domestic violence, environmental conditions and antenatal care. Our findings suggest that women in the Solomon Islands are exposed to various personal, behavioural, social and environmental risk factors during pregnancy that can impact birth outcomes, particularly LBW. We recommend further research should be redirected to look at the factors/themes identified in the interviews.

## Introduction

It is estimated that 140 million women are pregnant worldwide every year, with most of these pregnancies occurring in low-and middle-income countries (LMICs) [1]. Although pregnancy can be a rewarding experience and an exciting journey, this may not be the case for many women in LMICs due to their living conditions and exposure to risk factors that may adversely affect their pregnancies and birth outcomes [2–7]. Women's health during pregnancy in LMICs is largely influenced by socio-demographics, health, behavioural and environmental factors. Socio-demographic factors include maternal age, household income and education levels, and health factors include malaria infections and anaemia [8, 9]. Behavioural risk factors include substance use such as tobacco, alcohol and betel nut [10, 11], and environmental risks include the lack of access to antenatal care and poor sanitation [12, 13].

**Funding:** GP was supported with funding from the National Health and Medical Research Council Project and Investigator Grants #1099655 and #1173991 and the Research Council of Norway through its Centres of Excellence funding scheme #262700. GAT was supported with funding from the National Health and Medical Research Council Investigator Grant #1195716. LSK is a recipient of the Australia Award scholarship from the Department of Foreign Affairs. The funders had no role in study design, data collection and analysis, decision to publish, or preparation of the manuscript.

**Competing interests:** The authors have declared that no competing interests exist.

The Solomon Islands is located in the South Pacific region, an island country with a fragile subsistence-based economy that relies on foreign aid for health services and socio-economic development [14]. There are high rates of unemployment, illiteracy, and poverty [14], with more than 74% of the population living in rural areas as subsistence farmers [15]. Despite government efforts to ensure that health facility access is within an hour's reach, health service access and quality remain a problem, particularly for rural women [16]. Community health services and programs are often lacking in many rural communities due to geographical isolation and dispersed communities across the 992 scattered islands [16]. The number of health workers per population density is lower than that of most countries in the Asia Pacific region. Reports showed there were approximately 19 doctors, 145 nurses, and midwives per 100 000 of the population [17, 18]. Clean water supplies and sanitation are non-existent in more than half of the rural communities [19, 20], and substance use, especially alcohol, marijuana, and *kwaso* (illegally distilled alcohol), are prevalent among the population [21, 22]. The Solomon Islands is also a patriarchal society comprising various indigenous cultures, where men, as head of the family, own the land and possess inherited wealth, predisposing women to oppression and violence [23].

Low birth weight (LBW), defined as birth weight below 2500 grams is the most widely used perinatal benchmark for adverse birth outcomes globally [24]. LBW comprise 20% of global births (approximately 20 million per year), with 95% of these births occurring in LMICs [24]. The Solomon Islands has a prevalence for LBW of 10%, which places it within the minimal range compared to South-East Asian countries and Melanesian countries of the Pacific such as PNG and Vanuatu (between 10–17% prevalence) [2, 25–27]. However, this rate is elevated compared to some countries in the Polynesia region, with Tonga, Samoa and Tuvalu reporting LBW of well below 7% [2, 27]. In addition, LBW in the Solomon Islands was predominantly preterm births, with a recent hospital-based study reporting LBW accounted for 77% of neonatal deaths [28–30].

LBW can result from women's exposure to the various risk factors experienced during pregnancy as quantified by numerous studies in LMICs [2–7, 25, 27, 31–34]; however, limited studies have quantified the associated risk factors for LBW locally. Furthermore, no studies have been conducted in the Solomon Islands on women's lived experiences during pregnancy. This study aimed to understand the lived experience during the most recent pregnancy of women in the Solomon Islands who gave birth to LBW infants by exploring and describing their personal (socio-demographic and health), behavioural, social and environmental context during their pregnancy. The enquiry into the women's personal accounts will illuminate their unique experiences in their respective communities.

## Methods

### Study design and sampling

We employed a descriptive qualitative study using purposive sampling to recruit women with LBW infants to participate in one-on-one in-depth interviews. This methodology was deemed appropriate to explore women's personal stories and experiences during their recent pregnancies resulting in LBW babies [35, 36]. The consolidated criteria for reporting qualitative research (COREQ) checklist was used to ensure high-quality research reporting [36] (S1 File).

### Study setting and recruitment

Eighteen postnatal women aged 15 to 39 years were recruited from the special care nursery of the National Referral Hospital (NRH) in the Solomon Islands. NRH receives referral from all provinces. However, since it is located in Honiara on the Island of Guadalcanal, most of the

women were either referred from Honiara or rural Guadalcanal. The women were required to have given birth to LBW infants weighing less than 2500 grams, irrespective of gestational age, and within the last week before recruitment. It was also a requirement that the health of the mother and infant was stable. Women were required to be fluent in Pidgin English, the Solomon Islands lingua franca. The recruitment of participants was halted when data saturation was reflected in no new information emerging.

## Data collection

The lead author (LSK) conducted in-depth one on one semi-structured interviews with women within one week of delivery and before the infants were discharged from hospital. Due to travel restrictions resulting from the global COVID-19 pandemic, although it was originally planned to undertake face-to face interviews, all interviews were conducted via telephone. The processes were assisted by two local research assistants (RA) trained in ethical research processes. JT (RA1), a clinical nurse instructor in the paediatric and neonatal ward of the National Referral Hospital in the Solomon Islands, identified women meeting the inclusion criteria and invited them to participate in the telephone interviews after explaining the aims of the study, participant information sheet, and interview procedures and ensuring provision of secured private room for the women to sit during the interviews. LK (RA2), an academic at the University of South Pacific, Solomon Island Campus, supported the study by arranging and scheduling the telephone call processes and incentives for the women. Women received a pair of infant booties in appreciation of their time.

The interview schedule was assessed for face and content validity by an expert panel who have PhDs and expertise in clinical and public health research. It was then trialled with four Solomon Island women living in Australia to ensure the content was understandable and informed the research aim. We asked the women about their health, health behaviours, social and environmental conditions, access to antenatal care and their understanding of risk factors during pregnancy that may have contributed to having a LBW infant. Demographic data (age, residence, ethnicity, education level) were also collected (S1 Table). Prior to the interviews, a further three interviews were conducted with women of LBW infants in the Solomon Islands, to assess the interview schedule and the feasibility of the telephone interview method. Bracketing was undertaken through dialogue between the researchers to identify potential preconceived notions that might influence the data collection and analysis [37]. From this, a list of preconceived notions that might influence the data collection and analysis was written on a memo as a checklist guide to refer to during the data analysis.

The women who provided informed consent and permission for recording of the interview were placed in a secure private room for the interview. Confidentiality and privacy were strictly maintained for all participants throughout the interview process. RA1 (JT) escorted each participant into the room and assisted them in answering the telephone (if they were unfamiliar with the handset). Once comfortable, RA1 left and closed the door before the telephone interview commenced. The interviews ranged from 30 to 70 minutes, with an average time of 54 minutes.

## Data management and analysis

Data were de-identified by removing the participant names and replacing them with unique codes, that were uploaded to a password-protected database. All interviews were conducted in Pidgin English. Interviews were audio taped and field notes were taken. The lead author (LSK) transcribed all the interviews from Pidgin transcriptions directly into English, as she is fluent in both languages. Also, given that the Solomon Islands Pidgin creole is a derivative of the

English language combined with local languages, direct transcribing and translation were considered optimal as both languages are somewhat analogous in most terms. To ensure validity in the translation process, three translated scripts were reviewed by three colleagues fluent in Pidgin and English. The translated and transcribed data were uploaded into NVivo and then analysed using thematic analysis [38, 39]. This involved repeatedly reading through each interview to become immersed and familiar with its content and annotating meanings arising from the data; generating succinct labels (codes) to identify essential features of the data; collating data to identify significant broader patterns of meaning and potential themes; reviewing themes against the dataset to ensure they told the data's story; defining and naming themes; and weaving together the analytic narrative with quotes to illuminate themes [38, 39]. For validity and reliability, co-authors (JJ and GAT) who are experienced in qualitative research provided guidance and reviewed the generated themes with the lead author (LSK), leading to a consensus on the final themes.

## Ethical statement

Ethics approval was obtained from the Solomon Islands Health Research and Ethics Board of the Ministry of Health and Medical Services with ethics approval number HRE039/19. Reciprocal ethics was also granted by Curtin University Human Research Ethics Committee (HREC) with approval number HRE2020-0530. Informed consent was taken from all participants involved in this study. Written consent was taken from parents and guardians of mothers of infants of 18 years and below.

## Results

### Demographic characteristics of participants

A total of 18 women aged 15 to 39 years were recruited, 80% (n = 15) were in a union (permanent relationship), and 60% (n = 11) were from rural areas (S1 Table).

### Themes

Six major themes were identified: 1) health issues, 2) substance use, 3) diet and nutrition, 4) domestic violence, 5) environmental conditions, and 6) antenatal care.

### Health issues

Most women (n = 11) reported experiencing a range of physical health issues during their pregnancies, which they believed impacted their pregnancy outcomes, such as early birth and LBW infant. For example, a woman stated: *"I was bleeding the entire pregnancy, and that was why I had a small baby."[P8, 23-year-old]* Illnesses experienced during pregnancy included malaria and non-specific infections, antepartum haemorrhage (APH) and chest pain as attested by two of the women. *"At 4 months, I developed fever and shivering, got tested at the clinic and confirmed malaria positive. I believe I gave birth early due to the malaria infection."[P10,18-year-old]"I had prolonged chest pain [chronic] the entire pregnancy without any diagnosis of my illness. I delivered my baby at seven months."[P2,39-year-old]*

Women who experienced health problems during their pregnancy did not have the confidence to talk about these issues with the health staff, especially when it came to their sexual health. This was particularly so for young women aged less than 20 years who experienced symptoms suggestive of sexually transmitted infections (STIs) or urinary tract infections (UTIs) during their pregnancy. These young women did not inform the health staff, thereby no screening or treatment was provided. This was reflected by two 19-year-old women as

follows. *"I had experienced abnormal white foul pus coming out from my private parts with painful urination. I did not seek help from the clinic nurse."*[P15] *"I had yellow foul vaginal discharge during pregnancy. I did not tell the nurses. They [nurses] did not check me [vaginal swab] and I did not receive any treatments."*[P14]

Women from both rural and urban areas tended to engage in hard physical work due to their subsistence lifestyle. Five women reported physical stress, falls and accidents during their pregnancy, which they believed resulted in early labour and birth. For example, two of them described their experiences as follows. *"I fell twice at five months of pregnancy on my way to the garden and slipped over the slippery bush track while carrying a heavy load of garden produce to sell at the market. I barely had rest, from gardening and selling vegetable. From then on, I started to develop abdominal pains that led me to an early birth."*[P17,35-year-old, urban] *"The nurse told me the bleeding was because of hard labour of carrying a heavy load without much rest."*[P13,25-year-old, rural] Conversely five other women claimed they did not experience any major health problems during their entire pregnancy as reflected by a19-year-old. *"I was well during pregnancy except the regular morning sickness."*[P14]

## Substance use

Substance use was widespread among this population, especially betel nut, tobacco, marijuana and alcohol. Most women (n = 11) reported chewing betel nut and smoking locally grown or manufactured tobacco during pregnancy, despite knowing that these substances could have adverse health consequences. Some women (n = 4) even reported being heavy users, or not being able to stop using betel nut or tobacco during pregnancy. *"I am a heavy betel nut chewer; I chewed up to 7, 6, 5 nuts per day before and during my last pregnancy. I also smoked tobacco and savusavu roll [homegrown dried tobacco leaves]of 5 to 6 rolls per day for 3 years. Savusavu is powerful, can cause dizziness and bad feeling. I had to shower to feel better; then, I took one more roll. I think I had my baby very early because of this."*[P13,25-year-old] Some women (n = 3) denied or were unsure of betel nuts' potential harm to their fetus, especially among the rural women as highlighted by two of the women as follows: *"I don't think betel nut will affect my baby."*[P9, 25-year-old] *"I am not sure if betel nut is harmful for my unborn baby."*[P3,16-year-old]

A few women (n = 3) who were heavy chewers (3–6 betel nuts per day) believed betel nut helped them during their pregnancy, as it reduced the bad taste and gave them a good feeling and more energy, as expressed by two of the women: *"I could not stop chewing betel nut because it improved my taste and made me feel better."*[P6,20-year-old] *"I could not stop because betel nut gives me good feelings, energy to do things and keeps me awake."*[P18,18-year-old] Some women (n = 6) also reported witnessing pregnant women consuming other substances, including *kwaso* (home-made distilled alcohol), marijuana and beer during pregnancy. *"I have heard and seen other women have taken kwaso, marijuana and beer during pregnancy."*[P11,34-year-old]

One participant expressed how women use the substances during pregnancy to cope with stress. *"I knew of a neighbour who took too much alcohol and smoked marijuana during pregnancy and gave birth to a sick baby. She said she took those due to stress."*[P7,30-year-old] While substance use was quite prevalent among the women, several women (n = 7) abstained from these substances during pregnancy due to religious beliefs and knowledge of their harmful effects. *"As seventh day Adventists, we are not allowed to use betel nut and tobacco as it can cause problems with blood [anaemia]."*[P12,18-year-old] These women demonstrated some level of knowledge of the impact of substance use during pregnancy. *"I don't take any of these during pregnancy, I am too scared of them. Betel nut, tobacco and marijuana can lead to having a small baby."*[P7,30-year-old]

## Diet and nutrition

Knowledge about a nutritious diet was at times limited, especially among the less educated women (primary school or less), and those from rural areas. *"I don't know what a healthy diet is like."[P13,25-year-old]* Most participants (n = 14) reported eating at least three meals a day. *"I ate a roasted green banana [plantain] for breakfast, tea for lunch and sweet potatoes for dinner [3 meals]. I did not like to eat sausage."[P12,18-year-old]* However, their reported daily dietary intake during pregnancy tended to lack variety, often comprising starchy vegetables and carbo-hydrates (e.g., potatoes, rice) with limited quantities of protein. *"I usually eat potatoes, cassava, rice, cabbage like Amau [local cabbage], slippery cabbage [Ibika], and pakchoi [Chinese cabbage] and beans, tomato and eggplant. I think chicken is not healthy to eat during pregnan-cy."[P13,25-year-old]*

Although more than half of the participants (n = 10) reported local home-grown food to be a healthier option, some also believed that some meat and fish were not healthy and so these food types were not eaten during pregnancy. *"Sausage, chicken and meat are not healthy."[P12,18-year-old]* Many (n = 12) of the participants reported cultural food taboos, food restrictions, or personal preference for certain foods which contributed to their limited protein intake as reflected by two of the women. *"We are not allowed to eat crab, pig meat [pork] and the fish with big mouths during pregnancy."[P15,19-year-old]* *"Pregnant women are not allowed to eat ura [prawns], reef fish, shellfish and megapod eggs; they will cause problems during labour."[P14,19-year-old]*

For some women (n = 6) food security was impacted by environmental calamities, such as heavy rains and floods, which destroyed crops. *"Heavy rains and flood destroyed our food gar-den, and all our banana plants died which affected our food supply."[P12,18-year-old]* Also lim-ited finances reduced women's ability to purchase nutritious foods. *"The biggest challenge was to buy enough nutritious food and share with my household of extended family mem-bers."[P11,34-year-old]* Conversely, five women reported that they experienced no food restric-tions during their pregnancy and had an adequate food supply. *"I planted my own vegetables, so I have enough food with good nutrition."[P4,24-year-old]*

## Domestic violence

Almost half of the participants (n = 8) reported experiencing both physical and emotional vio-lence during their pregnancy from their partners or significant others. Many of them (n = 7) believed their exposure to violence had contributed to their child's early birth as expressed by two of the women. *"Every weekend, he would return drunk and beat me up. He beat me and kicked me on my back twice during pregnancy which led to the premature birth."[P15,19-year-old]* *"During the last pregnancy of my twin babies, he [husband] bashed me. It was a terrifying experience. As a result of this, I delivered my twins prematurely."[P7,30-year-old]* Although ten of the women did not experience domestic violence, most women understood the negative consequences of violence during pregnancy. *"Experiencing domestic violence during pregnancy can cause premature labour."[P15,19-year-old]*

## Environmental conditions

Some rural women (n = 5) described their dwellings and traditionally built homes as placing them at increased exposure to insects, particularly mosquitoes. *"We all lived in thatched sago palm-built house with one big open space where my family of seven lives [overcrowded]. It is not safe from pests and insects like mosquitoes."[P13,25-year-old]* Many of the women (n = 7) reported living in large extended families and believed that overcrowding increased their risk of infection. *We lived in a two-bed room house and built a small extension in the vicinity for 14*

*of us including relatives. It is very crowded with a high risk for spreading infection."[P11,34-year-old]*

Half of the women, living in rural and urban areas (n = 5), walked long distances to access water for drinking and domestic use. The water sources were often unsafe during rainy and drought seasons due to contamination. *"We used rainwater, stream and river nearby for drinking, cooking and domestic use. We must go downhill to reach the water sources. The stream and river get murky during rains or dries up during drought."[P14,19-year-old]* Two of the women claimed they had premature births due to stress from carrying heavy water containers. *"I think I had premature labour because I worked so hard every day, carrying large containers of water for household use from the river."[P16,32-year-old] "I carried water containers from the stream to the house uphill. One day I suddenly felt labour pains and gave birth early.[24 weeks]"[P10,18-year-old]*

Poor sanitation was pervasive, with a lack of proper toilet facilities throughout urban and rural areas. Some participants did not have toilet facilities and used bushes, creeks and the seaside to defecate. *"We do not have a proper toilet; we use the nearby bushes[chuckled]. In Wagina, we would use the seaside. We have not received any information on good toilets and sanitation."[P17,35-year-old] "We do not have a proper toilet like others in the community. We used a nearby creek as a toilet."[P15,19-year-old]* However, a few participants (n = 3) living in urban areas had inbuilt water and sanitary systems. *"I lived in a townhouse with an inbuilt modern toilet and water system."[P7,30-year-old]*

## Antenatal care

Women described several challenges related to their antenatal care. These included limited access to health services, limited provision of health information by health professionals, and their own inability to seek out health information. The ability to obtain antenatal health care was frequently impeded by access issues, such as distances to antenatal services, which was more common in rural areas, as well as geographical barriers (e.g., mountainous areas), extreme weather conditions (e.g., tropical rain), and unaffordable transportation costs as expressed by three of the women. *"Our clinic is very far from the village without road access [where we live] and took two hours by foot. I walked to get there, climbing hills and walking down the creek footpath which was very difficult."[P5,38-year-old] "My antenatal attendance depends on the weather. I would not go when the weather was bad."[P4,24-year-old]"The biggest challenge [to antenatal care] was the distance and travel costs. I would not go If I do not have the money for bus fare."[P12,18-year-old]*

Participants also nominated a range of barriers to obtaining optimal antenatal health care. These included the poor condition of health facilities, medicine shortages, no or inadequate health screening, long waiting times, and lack of professionalism among nursing staff. This was illustrated by participants who described the conditions women often experienced. *"Health clinic services [provision] should be improved; women should be checked properly and must be given malaria medicine [prophylaxis] during antenatal [care]."[P5,38-year-old]"The clinic needs proper clean beds for pregnant women to lay on. Also, the nurses always scold us, are slow in their performance, resulting in long waiting hours and some mothers just left."[P7,30-year-old]* Conversely, some participants (n = 7) reported better access to health facilities and appreciation of the antenatal cares service provided. *"My closest clinic is 50 meters away. I am satisfied with the services provided, and the information was adequate. The nurse checked my baby's position, gave health advice did a urine test and supplied the red medicine for blood [ferrous sulphate tablets]."[P1,20-year-old]*

## Discussion

### Summary of findings

Our study is the first to explore women's lived experiences during pregnancy in the Solomon Islands, providing an understanding of the perceived risks these women are exposed to during pregnancy that led to an early birth and LBW. Risk factors identified stemmed from the women's knowledge and experiences, which were described as personal (socio-demographic and health issues), behavioural (diet and nutrition and substance use) social (domestic violence) and environmental conditions. Although some women were conscious of the potential impact of these risk factors on pregnancy outcomes, many were unaware.

### Overview of findings

Commonly reported personal health issues experienced by the women during pregnancy included malaria, APH, STIs and UTIs, which are recognised as causes of early births and LBW [27, 40–42]. Malaria is endemic in the Solomon Islands [26], a known risk for LBW, affecting 125 million pregnant women globally [41, 43–45]. APH was another reported health issue, which can be triggered by infection, obstetric causes, or physical trauma [40]. Some women claimed that having bleeding during pregnancy was due to strenuous physical work. Others reported experiences of falls and accidents and perceived that these prompted the early onset of their labour, leading to early births. Similar findings have been reported in LMICs, where physical stress, falls, and trauma during pregnancy were recognised as causes of APH and early births [27, 42]. The lack of screening and treatment for STIs or UTIs was also a nominated issue, with women being unaware of the risks to their pregnancies. STIs and UTIs are well-established risk factors for LBW [46, 47], with a previous local study confirming that UTIs affected 59% of pregnant women and were linked to preterm births [30]. The women also reported not seeking help from health care providers for these STIs symptoms, a negative attitude to sexual health problems in the Solomon Islands, which has also been observed in a previous study [26]. Furthermore, STI screening, including vaginal swabbing is out of the scope of the current Solomon Island antenatal protocol, which is a shortfall in antenatal care [48, 49].

Our findings indicate that the interviewed women had limited knowledge of nutrition, reporting a diet high in starchy vegetables and active avoidance of meat protein (e.g., chicken and sausage) during their pregnancy. Local food taboos led to the further avoidance of protein, as certain fish were viewed as causing illness in newborns, a sickness referred to as *'fis'* [50]. Although there is limited local research on food taboos, studies in other LMICs have found them to be associated with suboptimal nutrition during pregnancy and adverse birth outcomes [51–53]. In the Solomon Island 41% of women of reproductive age experience anaemia, a condition mostly affecting women of low educational status and those living in rural areas, which further increases their risk of having preterm and LBW infants [26].

The women's diet was also impacted by food availability due to environmental factors, such as heavy rains and floods causing the destruction of crops and contributing to food insecurity [50, 54–56]. This is despite the introduction of the National Food Security, Food Safety and Nutrition Policy 2019–23 by the Ministry of Health and Medical Services (MHMS) [57]. Priority policy areas included better nutrition for vulnerable populations (e.g., women and children), strengthening of the food supply chain, awareness of safe and healthy food choices and promotion of health and nutrition. To date, it remains uncertain whether this has been well implemented at a consumer level.

Substance use (e.g., betel nut, tobacco, alcohol and marijuana) during pregnancy was also reported by our participants, with betel nut use found to be particularly frequent. Although

many of the women were aware of the impacts of these substances on pregnancy, there was a lack of knowledge and awareness, especially regarding betel nut use. There were also reports of betel nut addiction among some women who claimed to use betel nuts to increase energy levels and reduce bad tastes during pregnancy, which has also been reported in a Papua New Guinea (PNG) study [11]. Betel nut has been found to be associated with adverse birth outcomes, especially LBW and preterm birth, in several studies in PNG and Southeast Asia [10, 11, 13, 27, 58, 59]. Conversely, the women's use of tobacco, alcohol and marijuana during pregnancy was limited, with the women reporting some level of awareness of their harmful effects. The use of tobacco and alcohol during pregnancy and their effect on pregnancy and birth has been well documented [58, 60, 61]. Although evidence on marijuana use in pregnancy in LMICs is limited, studies in the United States have shown its use to be common among younger women and a potential risk factor for LBW and other adverse birth outcomes [61–63]. There were also reports that women tended to use these drugs to relieve stress, which has been reported in a previous Solomon Island study [23].

Domestic violence inflicted by an intimate partner or significant other was the social risk women reported. Almost half of the women in this study reported experiencing physical or emotional violence during their pregnancy. This is consistent with previous Solomon Islands studies, which have reported that up to 66% of women experience domestic violence from their intimate partner during pregnancy [23, 64]. Some of the women attested that being bashed or kicked in the back, led to early births, a serious consequence of violence during pregnancy [23, 64–67]. Despite the legislation of the Family Protection Act to protect families and promote safety and community education by the Solomon Islands government [64, 68], many of the women expressed a lack of social and community support, particularly in rural communities.

Environmental risks that women experienced included poor access to health facilities and lack of proper sanitation. The MHMS Role Delineation Policy underpins and safeguards affordable and accessible health care for its citizens, yet access to health care services remains a challenge [69]. It was evident that appropriate antenatal care was challenged by poor access, under-resourced health facilities, lack of staff professionalism and limited health information. Antenatal access was hampered by distances, geographical isolation, and extreme weather conditions, all governmental challenges in servicing this dispersed population [26]. Poor access to antenatal care impacts attendance, as reflected by previous local studies, which showed 85% of women have late antenatal bookings (trimesters 2 and 3), and 25% or less have only four antenatal visits [26, 70]. Poor access is a contextual risk for adverse pregnancy outcomes, which have been reported by other LMICs [12, 71]. Limiting health care, which is required during pregnancy [44, 45, 72], diminishes the ability to detect health risks and promote health services for better pregnancy outcomes. Women also expressed dissatisfaction with antenatal care due to not being treated with respect, and poor health facilities, also confirmed by one study from LMIC [71].

Poor water and sanitation remain pervasive in rural and urban areas of the Solomon Islands, even though the National Water and Sanitation Policy 12-year implementation plan was introduced in 2017 [73]. Poor water and sanitation poses a risk of vector and water-borne infections, as supported by a local study [26]. Although studies on sanitation and birth outcomes are limited, poor sanitation can threaten pregnancy due to infection [74, 75]. Furthermore, many of the women expressed that information on proper water and sanitation is limited in their community which may contribute to the lack of awareness of these risks.

## Study limitations and strength

This study provided an opportunity to talk with Solomon Island women from urban and rural areas, providing insights into their exposure to risk factors during their pregnancy. However,

due to the COVID-19 travel restrictions, the interviews were conducted by telephone, which may have inhibited the discussion and reduced the opportunity for the researcher to observe the women's body language. The women were mainly from Guadalcanal province and Honiara, and the findings were limited to these ethnic groups. Using a highly trained health professional as the interviewer can also be seen as a limitation as they may presume a greater level of participant knowledge than non-health professionals. Health professionals may themselves be influenced by aspects of care based on their prior experience, which governs the dialogue and interpretation. Participants may be less well-included to open up to trained health professionals who maintain their professional identity throughout the interview process compared to non-health professionals who are perceived as peers. On the other hand, the strength of the study was that the primary investigator is a female health professional, familiar with this type of participant, the clinical setting from which the women came, knowledgeable in the subject area, the local language of the participants (pidgin English) and translation processes.

## Conclusion

The study contributes to our understanding of the personal (socio-demographic and health), behavioural, social and environmental risk factors women who gave birth to LBW infants in the Solomon Islands experienced during pregnancy. The study identified women's knowledge and experience of potential risk factors which included health issues, diet and nutrition, substance use, domestic violence, inadequate antenatal care, and environmental conditions. We recommend further research should be redirected to look at the factors/themes identified in the interviews.

## Supporting information

**S1 Table. Women's characteristics.**
(DOCX)

**S1 File. COREQ checklist.**
(DOCX)

**S2 File. Table of quotes not included.**
(DOCX)

## Acknowledgments

We acknowledge all 18 women who have participated in the study. And we also acknowledge and thank our two research assistants, Registered Nurse Janet Tekatoha, Paediatric and Neonatal nurse instructor of the Special Care Nursery, National Referral Hospital, Solomon Islands, for recruiting and preparing participants for the interviews. We also acknowledge Lillian Kaforau, an academic at the University of South Pacific Solomon Island campus, for support with logistics.

## Author Contributions

**Conceptualization:** Lydia S. Kaforau, Gizachew A. Tessema, Hugo Bugoro, Gavin Pereira, Jonine Jancey.

**Data curation:** Lydia S. Kaforau.

**Formal analysis:** Lydia S. Kaforau, Jonine Jancey.

**Investigation:** Lydia S. Kaforau.

**Methodology:** Lydia S. Kaforau, Gizachew A. Tessema, Jonine Jancey.

**Project administration:** Lydia S. Kaforau.

**Software:** Lydia S. Kaforau.

**Supervision:** Gizachew A. Tessema, Hugo Bugoro, Gavin Pereira, Jonine Jancey.

**Validation:** Lydia S. Kaforau, Gizachew A. Tessema, Jonine Jancey.

**Writing – original draft:** Lydia S. Kaforau.

**Writing – review & editing:** Lydia S. Kaforau, Gizachew A. Tessema, Hugo Bugoro, Gavin Pereira, Jonine Jancey.

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
