## [Decision Letter · Decision Letter 0]

7 Sep 2022

PGPH-D-22-01214

Lived experiences of women with low birth weight infants in the Solomon Islands: a descriptive qualitative study.

Dear Dr. Kaforau,

Thank you for submitting your manuscript to PLOS Global Public Health. After careful consideration, we feel that it has merit but does not fully meet PLOS Global Public Health’s publication criteria as it currently stands. Therefore, we invite you to submit a revised version of the manuscript that addresses the points raised during the review process.

We look forward to receiving your revised manuscript.

Kind regards,

Nicola L. Hawley

Academic Editor

Journal Requirements:

1. Note from Staff Editor Katrien Janin (kjanin@plos.org): PLOS considers qualitative and mixed-methods studies for publication. We recommend that authors use the COREQ checklist, or other relevant checklists listed by the Equator Network, such as the SRQR, to ensure complete reporting (https://journals.plos.org/globalpublichealth/s/submission-guidelines#loc-qualitative-research). In general, we would expect qualitative studies to include the following: 1) defined objectives or research questions; 2) description of the sampling strategy, including rationale for the recruitment method, participant inclusion/exclusion criteria and the number of participants recruited; 3) detailed reporting of the data collection procedures; 4) data analysis procedures described in sufficient detail to enable replication; 5) a discussion of potential sources of bias; and 6) a discussion of limitations. In your role as Academic Editor, we appreciate your consideration of whether the manuscript meets reporting standards in the field, in addition to the journal’s other publication criteria (https://journals.plos.org/globalpublichealth/s/criteria-for-publication). Please feel free to email me to discuss the work further.

Additional Editor Comments (if provided):

Reviewers' comments:

Reviewer's Responses to Questions

**Comments to the Author**

1. Does this manuscript meet PLOS Global Public Health’s publication criteria? Is the manuscript technically sound, and do the data support the conclusions? The manuscript must describe methodologically and ethically rigorous research with conclusions that are appropriately drawn based on the data presented.

Reviewer #1: Yes

Reviewer #2: Yes

2. Has the statistical analysis been performed appropriately and rigorously?

Reviewer #1: N/A

Reviewer #2: N/A

3. Have the authors made all data underlying the findings in their manuscript fully available (please refer to the Data Availability Statement at the start of the manuscript PDF file)?

Reviewer #1: Yes

Reviewer #2: No

4. Is the manuscript presented in an intelligible fashion and written in standard English?

Reviewer #1: Yes

Reviewer #2: No

5. Review Comments to the Author

Reviewer #1: 1. Summary of the Research

a. Research question, claims, and conclusions of the study

Thank you for providing this opportunity to read and provide feedback on this article. This research article reports on the lived experience of a purposive sample of 18 Solomon Islands women who had recently given birth to a baby of low birth weight, through the usage of a qualitative descriptive methodology. The interview protocol was reviewed by experts and administered in the native language to the women participants. Due to travel restrictions, interviews were conducted via telephone. Recordings were translated into English and NVivo used for analysis. Six thematic areas were identified; health issues, substance use, diet and nutrition, domestic violence, environmental conditions and antenatal care. The authors’ findings submit that the women were exposed to different personal, behavioral, social and environmental risk factors that can affect birth outcomes with specific mention of LBW

b. Context for how this research fits within the existing literature

The authors make comparison to other published studies in other regions of the world such as Asia and the USA, and contextualizes their findings with other published articles on the thematic area in the region where possible (example of PNG). The authors utilize reports produced by the Solomon Islands to contextualize their findings such as reports on domestic violence. They seek to fulfill a gap in the published literature on the lived experiences of women in the Solomon Islands during their pregnancy.

c. Discuss the manuscripts strengths and weaknesses and your overall recommendations

This is very interesting report that sheds light on the experiences of pregnant women in the Solomon Islands. The authors provides a sound rationale for the methodology chosen and the approach to data collection and analysis. The inability to conduct the interviews in person removed the ability to respond to facial expressions and non-verbal cues. With some revision, the report would be a lovely addition to the body of literature in this space. The recommended changes are

• include a section on setting in the background or methodology to assist the reader to better understand the situation in the Solomon Islands – the location, development status, ratio of health workers to population, percentage of births that are LBW etc

• Include a detailed discussion that compares and situates that the thematic areas within the data available on the Solomon Islands

• Consider including the statistics referred to instead of only including the citation, to provide the reader with specific information.

• The Solomon Islands is a Melanesian country culturally but is compared to data from countries in the Polynesia region, perhaps comparison can also be made to other Melanesian countries.

• Introduce each of the acronyms

• Review the grammar (there were several mismatches between the noun and verb)

• Provide the range of the length of time of interviews instead of only the average

• Format to highlight long quotes

• The usage of the words, many, several, most, some, should be companied by a number.

Reviewer #2: Summary of review: Overall, this paper contributes important qualitative information on LBW in a region that has little data. There were grammatical (including run-on sentences) and spacing errors, so authors should perform a close edit of the paper overall. The authors used the COREQ framework, which ensured that the methods and reporting were high quality. The results are described well with illuminating quotes and are important findings that could direct future research on the topic. The discussion was also clear and could benefit if the authors could expand on other public health policies in the country related to the themes they identified. To meet the open data sharing requirement of the journal, authors could also produce a table with other quotes on the themes that were not included in the results.

Abstract

-Should be comma: “qualitative descriptive”

-Could be nice to remind the reader what the themes were describing in this sentence under Findings.

Introduction

-Sometimes “socio-demographic” and other times “sociodemographic” so please choose one way of spelling.

-spacing error between “such as” and “tobacco”

-another spacing error, “poverty[14]”

-Could state which countries are below 7% LBW for readers

-“Furthermore, no studies have”

Methods

-“The recruitment of participants was halted when data saturation was reflected in no new information emerging.” How did the authors define saturation and know when they reached it?

-The interview schedule was assessed by an expert panel. Could the authors please describe this in further detail, such as the qualifications of who was on the panel.

-“The interview questions asked about”, sound colloquial and makes sentences difficult to read.

-Could the authors be more specific on how they used bracketing as a method? What were the preconceived notions?

Results

-Quotes were integrated nicely into the reporting of the results and painted a vivid picture of the experiences of the women in this study

Discussion

-Is there any scholarly work done on policy or public health education in the Solomon Islands on themes identified in this paper, such as substance use during pregnancy or domestic violence? This could be an important discussion point of the applications of this research. Expanded discussion on policies or education like the authors did of the Family Protection Act, could provide important political background and highlight potential areas to target for interventions.

-In the limitations, although it could be a perceived as a strength to have the interviewer be a health professional, have the authors considered whether this was also a limitation? Could there be a reporting bias is interviewees give answers that they think the doctor would like to hear? It doesn’t appear so, given that the participant spoke freely about some unhealthy behaviors of substance abuse during pregnancy, but it could be nice for authors to explore the provider/interviewee dynamic more.

6. PLOS authors have the option to publish the peer review history of their article (what does this mean?). If published, this will include your full peer review and any attached files.

**Do you want your identity to be public for this peer review?** For information about this choice, including consent withdrawal, please see our Privacy Policy.

Reviewer #1: No

Reviewer #2: No

---

## [Decision Letter · Decision Letter 1]

11 Nov 2022

Lived experiences of women with low birth weight infants in the Solomon Islands: a descriptive qualitative study.

PGPH-D-22-01214R1

Dear Mrs Kaforau,

We are pleased to inform you that your manuscript 'Lived experiences of women with low birth weight infants in the Solomon Islands: a descriptive qualitative study.' has been provisionally accepted for publication in PLOS Global Public Health.

Best regards,

Nicola L. Hawley

Academic Editor

Reviewer Comments (if any, and for reference):

Reviewer's Responses to Questions

**Comments to the Author**

1. If the authors have adequately addressed your comments raised in a previous round of review and you feel that this manuscript is now acceptable for publication, you may indicate that here to bypass the “Comments to the Author” section, enter your conflict of interest statement in the “Confidential to Editor” section, and submit your "Accept" recommendation.

Reviewer #1: All comments have been addressed

Reviewer #2: All comments have been addressed

2. Does this manuscript meet PLOS Global Public Health’s publication criteria? Is the manuscript technically sound, and do the data support the conclusions? The manuscript must describe methodologically and ethically rigorous research with conclusions that are appropriately drawn based on the data presented.

Reviewer #1: Yes

Reviewer #2: Yes

3. Has the statistical analysis been performed appropriately and rigorously?

Reviewer #1: N/A

Reviewer #2: N/A

4. Have the authors made all data underlying the findings in their manuscript fully available (please refer to the Data Availability Statement at the start of the manuscript PDF file)?

Reviewer #1: Yes

Reviewer #2: Yes

5. Is the manuscript presented in an intelligible fashion and written in standard English?

Reviewer #1: Yes

Reviewer #2: Yes

6. Review Comments to the Author

Reviewer #1: This paper will be very useful to researchers, academics and clinicians, especially in light of the lack of available literature on the topic in the geographic region. The authors have provided detailed responses to reviewer comments. They have made the necessary amendments within the manuscript and made the quotes from interviewees available in the supplementary data. As this is a qualitative study, basic demographic information on the interviewees has been provided, but further statistical analysis was not done. This is the explanation for the N/A response to the question on statistical analysis.

Reviewer #2: Authors have addressed all comments.

7. PLOS authors have the option to publish the peer review history of their article (what does this mean?). If published, this will include your full peer review and any attached files.

**Do you want your identity to be public for this peer review?** For information about this choice, including consent withdrawal, please see our Privacy Policy.

Reviewer #1: No

Reviewer #2: No
